# The Influence of Herbicides to Marine Organisms *Aliivibrio fischeri* and *Artemia salina*

**DOI:** 10.3390/toxics9110275

**Published:** 2021-10-21

**Authors:** Radek Vurm, Lucia Tajnaiová, Jana Kofroňová

**Affiliations:** Faculty of Environmental Technology, Department of Environmental Chemistry, UCT Prague, Technická 5, 166 28 Prague, Czech Republic; kofronoj@vscht.cz

**Keywords:** *Aliivibrio fischeri*, *Artemia salina*, ecotoxicology, herbicides, glyphosate

## Abstract

The aim of this work was to determine the toxic effect of the most used herbicides on marine organisms, the bacterium *Aliivibrio fischeri*, and the crustacean *Artemia salina*. The effect of these substances was evaluated using a luminescent bacterial test and an ecotoxicity test. The results showed that half maximal inhibitory concentration for *A. fischeri* is as follows: _15min_IC_50_ (Roundup^®^ Classic Pro) = 236 μg·L^−1^, _15min_IC_50_ (Kaput^®^ Premium) = 2475 μg·L^−1^, _15min_IC_50_ (Banvel^®^ 480 S) = 2637 μg·L^−1^, _15min_IC_50_ (Lontrel 300) = 7596 μg·L^−1^, _15min_IC_50_ (Finalsan^®^) = 64 μg·L^−1^, _15min_IC_50_ (glyphosate) = 7934 μg·L^−1^, _15min_IC_50_ (dicamba) = 15,937 μg·L^−1^, _15min_IC_50_ (clopyralid) = 10,417 μg·L^−1^, _15min_IC_50_ (nonanoic acid) = 16,040 μg·L^−1^. Median lethal concentrations for *A. salina* were determined as follows: LC_50_ (Roundup^®^ Classic Pro) = 18 μg·L^−1^, LC_50_ (Kaput^®^ Premium) = 19 μg·L^−1^, LC_50_ (Banvel^®^ 480 S) = 2519 μg·L^−1^, LC_50_ (Lontrel 300) = 1796 μg·L^−1^, LC_50_ (Finalsan^®^) = 100 μg·L^−1^, LC_50_ (glyphosate) = 811 μg·L^−1^, LC_50_ (dicamba) = 3705 μg·L^−1^, LC_50_ (clopyralid) = 2800 μg·L^−1^, LC_50_ (nonanoic acid) = 7493 μg·L^−1^. These findings indicate the need to monitor the herbicides used for all environmental compartments.

## 1. Introduction

Currently, it is no longer possible to practice a modern form of agriculture without using plant protection products. Identification and comparison of their environmental impacts have a significant role in the protection of the environment and humans. How herbicides affect life also depends on their ability to move in the environment. The most important factors influencing their distribution include air temperature, light insensitivity [1], enzyme activity [2], rainfall, runoff, field position [3], wind speed, soil type, soil moisture, dew effect [4], and growth weed phase [5]. In this way, plant protection products can get from fields, turf grass, and residential areas to rivers and seas. Herbicide run-off may pose a potential threat to non-target organisms such as changes in macroinvertebrate communities [6,7] or in physical condition of amphibians [8]. Repeated application and high doses increase the risk of more available chemical for runoff [9]. Herbicides can have indirect toxic effects on fish due to destruction of their natural habitats or reduced amount of dissolved oxygen [10]. The percentage of herbicide that has been applied to the fields and subsequently reached the surface runoff can range from 0.05% to 43.5% depending on several factors, such as the solubility of the herbicide [3]. Other factors include, for example, wind speed [11], boom height [12], distance from susceptible crop [13], and spray particle size [14]. Other factors such as rainfall, greater initial soil water content, and crop residue cover can also affect the concentration of herbicide outside the application area [15].

From polluted rivers and lakes, residues of the original compounds and their transformation products can be leached into aquifers [16]. Contamination with plant protection products can be minimized by management practice and techniques based on specific local conditions [17,18,19]. Plant protection products usually consist of active substance and additional “inactive” ingredients known as adjuvants (surfactants, emulsifiers, solvents, etc.), which are crucial and may have impacts on the environment [20,21,22]. Many of them are considered high volume chemicals and are usually mixed in the formulation, which may show moderately different behavior in the environment compared to the single compound [23]. Little background information about the other ingredients is usually known but they may have an effect on herbicidal activity [24], leaf coverage, and amount of active substance needed [25] on mobility of different compounds in soil [26,27,28], and so on. It is considered that surfactants are partly dissipated through sunlight or metabolism after the application [21] but they may have an influence even after a long time, such as an increased desorption of herbicides [29]. At the same time, they may be the cause of observed lethal or sublethal effects [30]. Some common ingredients used in commercial plant protection products were tested on several organisms and their effects, such as high toxicity on honey bees (N-methyl-2-pyrrolidone) [31], decreased activity on two-spotted spider mites (trisiloxane) [25], or estrogenic activity in male rainbow trout (alkylphenols) [32] were evaluated. Inactive compounds may be easily degraded but the products may be more toxic; for instance, alkylphenoxy ethoxylates are generally less toxic than products of their degradation, e.g., nonyl and octylphenols [21,23]. These findings indicate the need for testing the formulations, which include the active substances.

Glyphosate (N-(phosphonomethyl) glycine) is a nonselective, systemic herbicide with a short half-life, which decomposes rapidly in water into metabolites, such as aminomethylphosphonic acid (AMPA) [33,34]. Herbicide degradation processes are another important aspect to consider when assessing the environment. Glyphosate and AMPA can be found in surface water and groundwater [35,36], and even in salt water [37]. Levels of glyphosate in the Baltic Sea were determined to be between 0.42 and 1.22 ng·L^−1^, whereas a new methodology for the determination of glyphosate and AMPA in salt water were established [38]. The effects of herbicide degradation products on marine organisms have not yet been sufficiently investigated. According to Matozzo et al., AMPA can affect cellular and biochemical parameters in mussels *Mytilus galloprovincialis* [39]. Synergistic effects of glyphosate and AMPA on the same organism were also observed [40]. Mostly, it is AMPA that can negatively affect non-target organisms in the environment; examples of such effects are morphological changes in the aquatic plant duckweed *Lemna minor* and toxicity to green algae *Desmodesmus subspicatus* [41]. Chronic exposure to glyphosate along with other 2,4-D herbicides also affects the growth change and swimming activity of *Boana faber* and *Leptodactylus latrans* tadpoles. Furthermore, erythrocyte abnormalities and damage in the eating and intestinal areas were reported [42]. Cases of hepatotoxic effects of glyphosate on juvenile common carps [43] and mice [44] were also described.

Dicamba (3,6-dichoro-2-methoxybenzoic acid) is a synthetic auxin type of herbicide [45]. Synthetic auxins mimic the activity of natural phytohormone auxins, causing the overreaction that may lead to excessive growth, deformation, and plant exhaustion. They may be specious-selective, which is used for application on dipots [46]. Plants’s ability to control the levels of synthetic auxins is worse than that of natural ones, which increases their toxicity [47]. Published literature even reports the ability of auxins to interact and have toxic effects to lipid biomembranes [48]. The most famous natural auxin is indole-3-acetic acid (IAA) that may be found in bacteria, fungi, algae [48], and in animals such as mammals [49]. Many bacteria are able to synthesize IAA through different pathways [50] and some microorganisms are even capable of degrading dicamba [51,52].

According to their specific mode of action, their toxicity is not so significant on some model organisms (such as bacteria or crustaceans), especially in time- and environmental concentration-dependent toxicity tests [53,54,55]. Recently, there have been growing concerns about the presence of herbicides in drinking water affecting non-target organisms. According to Filkowski et al., the presence of dicamba in water supplies can pose a potential hazard to the genetic material of exposed living organisms [56]. Recent research suggests that dicamba should be considered a potential endocrine disruptor [57].

Clopyralid (3,6-dichloro-2-pyridinecyrboxylic acid) is a selective and growth regulator herbicide that is highly soluble in water [58]. It is often found in drinking water [59], even at higher concentrations than the permitted value (0.1 μg·L^−1^) for this pesticide [60]. Clopyralid residues were also found in several crops grown on soil contaminated with this herbicide [61] and has also been detected in processed cereals, such as wheat or barley bran; great attention is paid to its accumulation in compost [62,63,64,65].

Nonanoic acid or pelargonic acid is a long-chain fatty acid naturally occurring in numerous fruits and vegetables [66]. It is used in organic synthesis [67]. It is considered a contact herbicide: If applied to a leaf, it attacks and destroys plant cell membranes and causes tissue dehydration. The herbicidal effect of this acid is rapid, non-selective and broad-spectrum, leading to necrotic lesions on plants [68]. Its inhibitory effect on *Microcystis aeruginosa* growth is reported as EC_50_ = 0.5 mg·L^−1^ [69].

*Aliivibrio fischeri* is a Gram-negative marine bioluminescent bacterium that is widely used in toxicity tests [70], for example, for testing toxicity in the sediment environment [71]. *A. fischeri* is also used as a biomodel selected to obtain a mathematical model for predicting ecotoxicity [72]. Bioluminescence inhibition assays show a good correlation with other toxicity tests, including crustaceans [73]. The basis of the biochemical mechanism of *A. fischeri* luminescence is the reduction of flavin mononucleotide in reaction with the reduced form of nicotinamide adenine dinucleotide phosphate in the presence of flavin reductase enzyme [74]. *A. fischeri* photobacteria are also used to determine interactive toxic effects. According to Yang et al., the toxicity of heavy metals (Zn^2+^, Cu^2+^, Cd^2+^) to *A. fischeri* increases with reaction time, while the toxicity of organic substances (phenol, benzoic acid, p-hydroxybenzoic acid, nitrobenzene, and benzene) varies in different reaction times. This difference is due to the fact that for metals the rate of inhibition of *A. fischeri* bioluminescence is significantly higher than the relative rate of cell death, while for organic substances the rate of cell death is similar to bioluminescence inhibition [75].

The brine shrimp *Artemia* is widely used in biological studies [76], research, and toxicology [77] because it is easy to culture [78,79] and due to the good commercial availability of dried cysts [77]. It also serves as a model organism for assessment of the aquatic toxicity [80]. *Artemia* is very important because it is a part of the food chain [81]. *Artemia* spp. are crustaceans that can inhabit chloride, sulfate, and carbonate waters [82], and they are tolerant to variable oxygen levels [83] and salinity [76]. Determining the effect of toxic substances on *Artemia salina* may be affected by their age [84]. *Artemia* accumulates mercury, copper, and chromium in their body, with observed mortality rates in the order Cr > Hg > Cu [85]. Food colorants can affect mortality, mobility, and phototactic reactions in nauplii [86]. Another substance that has a negative effect on *A. salina* are silver nanoparticles (30–40 nm) in nanomolar concentrations (2–12 nM) which cause increased mortality, intestinal aggregation, the frequency of apoptotic cells, and DNA damage in nauplii in direct correlation with increasing concentrations of the toxicant [87].

The main aim of this work was to determine the acute toxicity of selected herbicides and their active substances to two marine organisms, bioluminescent bacteria *A. fischeri*, and nauplii of crustacea *A. salina*. The information obtained may be useful for assessing the risk of pesticide use and for further setting up tests. In addition, the saltwater species toxicity database is still insufficient. For example, the risk of many agricultural chemicals escaping into salty marshes cannot be properly assessed until further data on the toxicity of these chemicals to saltwater species have been tested [88]. Based on the measured data, we determined half maximal inhibitory concentration (IC_50_) for the marine bacterium *A. fischeri* and lethal concentration, which kills 50% of tested animals (LC_50_) for crustaceans *A. salina* for the following herbicides and their active substances: Roundup^®^ Classic Pro, Kaput^®^ Premium, Banvel^®^ 480 S, Lontrel 300, Finalsan^®^, glyphosate, dicamba, clopyralid, and nonanoic acid. Herbicides were selected on the basis of the best-selling and most widely available plant protection products on the Czech market. In addition, glyphosate is the most used herbicide worldwide [89,90,91,92]. For the purpose of comparison, we have selected two major products on the market (Roundup^®^ Classic Pro, Kaput^®^ Premium) that share the same active ingredient (glyphosate).

## 2. Materials and Methods

### 2.1. Chemicals

Five different chemical substances were tested: Roundup^®^ Classic Pro (manufactured by Monsanto Canada, Ottawa, ON, Canada), Kaput^®^ Premium (manufactured by Nohel Gardel, Dobříš, Czech Republic), Banvel^®^ 480 S (manufactured by Syngenta, Basel, Switzerland), Lontrel 300 (supplied by AgroBio Opava, Brumovice, Czech Republic), and Finalsan^®^ (manufactured by Neudorff, Brno, Czech Republic). The first two are glyphosate-based herbicides and both contain 28.85% *w*/*v* of glyphosate. Roundup^®^ Classic Pro further includes the additive surfactant ether alkylamine ethoxylate (6%), and small amounts of other chemicals. Kaput^®^ Premium contains the following formulation ingredients: N-(phosphomethyl) glycine (41.5% *w*/*v*), amine salt of phosphate ester (5–15% *w*/*v*), and other substances. Banvel^®^ 480 S contains 3,6-dichloro-o-anisic acid, combined with dimethylamine (1:1) (dicamba-dimethylammonium) (30–50% *w*/*v*) as an active substance. The fourth herbicide studied in this work was Lontrel 300. Lontrel contains a mixture of clopyralid monoethanolamine salt (35% *w*/*v*), and alkylphenol alkoxylate (less than 5% *w*/*v*). The last investigated chemical substance with herbicidal effect was Finalsan^®^. Finalsan^®^ consists of 18.67% *w*/*v* nonanoic acid, and 4% *w*/*v* propan-2-ol. In addition, four pure compounds were tested, which were used as active substances in tested herbicides: glyphosate (analytical standard, Sigma–Aldrich, St. Louis, MO, USA), dicamba (analytical standard, Sigma–Aldrich), clopyralid (analytical standard, Sigma–Aldrich), and nonanoic acid (purity > 97%, Sigma–Aldrich).

For *V. fischeri* test, a reactivation solution for bacteria (LCK 482, Hach Lange GmbH, Berlin, Germany) was used. Dimethyl sulfoxide (DMSO) (purity 99%, Penta) was used to increase the solubility of glyphosate, dicamba, and clopyralid. Sodium chloride (purity > 99.9%, Lachner) was used as diluent. Potassium dichromate (purity >99%, Sigma–Aldrich), and zinc sulfate heptahydrate (purity 99%, Penta) were used as standards to validate the method.

### 2.2. Organisms

Two saltwater organisms were chosen for this experiment. The first one were bacteria *Aliivibrio fischeri*, LCK 482—strain 20,275, Hach Lange GmbH, Germany. The second one were crustaceans *Artemia salina*, salt gill cysts, Easyfish, Czech Republic.

### 2.3. Experimental Design

The experimental design of *A. fischeri* test was carried out according the following standard: ISO 11348-2: Water quality. Determination of the inhibitory effect of water samples on the light emission of *Vibrio fischeri* (Luminescent bacteria test). Part 2: Method using liquid-dried bacteria [93].

The test substances were diluted in sodium chloride solution (2%) and measured (see Appendix A). The concentrations of the herbicide components were calculated on the basis of the information in safety data sheets. The concentration range was selected based on a preliminary screening assay that was between 20 and 200,000 μg·L^−1^.

DMSO was used to increase the solubility of the test substances. DMSO was used in every test with glyphosate, clopyralid, and dicamba. Control measurements showed no toxicity at used concentrations. Test samples, controls, standards, and reactivation solution were tempered to 15 °C in a thermoblock (TS 15, Meopta, Přerov, Czech Republic). *A. fischeri* bacteria were stored in a freezer (Indesit, Fabriano, Italy) at −18 °C. Before the test, they were left in a water bath at 18 ± 2 °C for 2 min, after thawing they were mixed with 500 μL of reactivation solution and homogenized. They were then mixed with 11.5 mL reactivation solution, shaken, and tempered to 15 °C. After adaptation of the bacteria in test tubes (0.5 mL per tube), luminescence was measured using a luminometer (LM O2 Z, Meopta, Czech Republic). After each measurement, 0.5 mL of the test substance, sodium chloride solution, or standard solution was added to the test tubes. The time interval for luminescence measurement was 30 s. Luminescence inhibition was measured after 15 min and 30 min. All samples were measured in triplicate. From the measured values, a correction factor was calculated, from which the corrected luminescence value for individual tubes was determined. Subsequently, the luminescence inhibition value was calculated. The equations used for the calculation are given in the standard ISO 11348-2.

In the crustacean bioassay test, all tested substances were diluted in a sodium chloride solution and measured (see Appendix A).

Dried *Artemia* cysts were hatched into nauplii by the following process: a 3% sodium chloride solution was poured into a cylindrical vessel intended for hatching crustaceans (JBL Artemio Set, JBL, Neuhofen, Germany) and *Artemia* cysts were added. The vessel was sealed, placed into a cultivator (Q-cell, Poland) at 24 ± 1 °C, and continuously aerated (aerator JBL, Germany). Lighting conditions were secured by a LED lamp (continuous lighting, 2000 lx). After 24 h, the hatched nauplii were separated using the bottom tap of the vessel and transferred to a glass aquarium with 3% sodium chloride solution. Subsequently, 10 mL of the test solution of the substance, control (3% sodium chloride solution), or potassium dichromate standard was placed in Petri dishes (diameter 60 mm). Ten pieces of nauplii were transferred to each dish using a Pasteur pipette. The Petri dishes were closed, transferred to a cultivator at a set temperature of 24 ± 1 °C. The lighting was set at 2000 lx (12 h light: 12 h dark). During the test, the Petri dishes were not aerated and the *Artemia* were not fed. After 24 h, the numbers of living and dead *Artemia* were checked. The test was considered valid if the mortality in the control assays did not exceed 10%. Mortality was calculated according to the equation:(1)$$M_{t}=\frac{N_{M}}{N_{0}}\times 100,$$
where *M_t_* is the mortality at a given time (%), *N_M_* is the average mortality of individuals at a given concentration of a substance or control, *N*_0_ is the number of individuals in each Petri dish at the beginning of the test. All measurements were made in four-fold determination; controls were in six-fold determination.

### 2.4. Statistical Analysis

The probit analysis was used to evaluate ecotoxicological tests. The probit-log (concentration) regression model was used to calculate slopes and intercepts (see Appendix A). The Microsoft Excel software was used for the calculation of the LC_50_, LC_90_ values and fiducial confidence intervals at 0.05 level of significance [94]. For both tests, the plots were constructed that show differences between the toxicity of herbicides and active substances using the Microsoft Excel software.

## 3. Results and Discussion

### 3.1. Luminescent Bacteria Test

Table 1 shows the IC_50_ and IC_90_ values for all tested herbicides and active substances. The half maximal inhibitory concentration for Roundup^®^ Classic Pro was 236 μg·L^−1^ after 15 min and 243 μg·L^−1^ after 30 min since the measuring started. The value of _15min_IC_50_ for Kaput^®^ Premium was found to be 2475 μg·L^−1^, the value of _30min_IC_50_ was measured to be 2598 μg·L^−1^. This indicates that although both of these herbicides have the same active ingredient, Roundup^®^ Classic Pro is more than 10.4 times more toxic to *A. fischeri* than Kaput^®^ Premium.

The toxicity values of Roundup^®^ Classic Pro and Kaput^®^ Premium are significantly different from those of glyphosate. In this case, the values were determined to be _15min_IC_50_ = 7934 μg·L^−1^ and _30min_IC_50_ = 2928 μg·L^−1^. IC_50_ values after 15 and 30 min of exposure did not show significant differences for Roundup^®^ Classic Pro; however, in the case of glyphosate, there is more than 2.7-fold difference in the inhibition values. Some excipients may increase the effects of pesticides, so it is appropriate to perform toxicity tests on a formulation, such as Roundup, rather than on the pure substance itself [95]. In addition to glyphosate, the tested Roundup^®^ Classic Pro also includes ether alkylamine ethoxylate, which is classified as toxic to aquatic organisms [96] with long-term effects. On the other hand, Kaput^®^ Premium does not contain such a substance [96]. An earlier study found that the EC_50_ value for glyphosate in the form of an isopropylamine salt is 36,900 μg·L^−1^ [97], which is 4.6 times higher than our measurement showed. The obtained results for glyphosate show higher toxicity as the results from previous studies, which indicate _15min_IC_50_ = 18,230 [98], _15min_IC_50_ = 43,800 [99], _30min_IC_50_ = 21,250 [98], and _30min_IC_50_ = 44,200 μg·L^−1^ [99].

Inhibitory concentrations were determined to be _15min_IC_50_ = 2637 and _30min_IC_50_ = 2286 μg·L^−1^ for Banvel^®^ 480 S and _15min_IC_50_ = 15,937 and _30min_IC_50_ = 9220 μg·L^−1^ for dicamba. In the acute toxicity tests of Banvel^®^ 480 S, reduced bioluminescence was observed at a concentration as low as 1000 μg·L^−1^, at which point the inhibition was 21.9%. Regarding dicamba, its active substance, 22.2% inhibition was achieved at the concentration of 5000 μg·L^−1^. As with glyphosate, the difference in the inhibitory effect between 15 and 30 min exposures was more pronounced with dicamba. According to the CLP regulation [96], in Banvel herbicide formulation only dicamba is classified as harmful to the aquatic environment. However, bacteria can also be negatively affected by other solvent substances such as acetone, ethanol or various organochlorine solvents [100]. The results obtained for dicamba are lower than previously published study results, where the IC_50_ value was determined to be 56,620 μg·L^−1^ at a 15 min exposure to dicamba and 36,250 μg·L^−1^ at the 30 min exposure. However, in their study, Westlund et al. point out the underestimation of toxicity results using the traditional 30 min evaluation of bioluminescence tests on *A. fischeri* [53]. Although, in the case of a chronic test lasting 20 h, no toxicity was observed for dicamba. Increased toxicity in the chronic test was observed with the fungicides climbazole and propiconazole, and the herbicides atrazine, irgarol, mecoprop, and diuron [53].

The value of _15min_IC_50_ for Lontrel 300 was found to be 7596 μg·L^−1^, the value of _30min_IC_50_ was measured to be less than 8740 μg·L^−1^. For Lontrel 300, lower toxicity was observed after 15 min of exposure, the opposite situation occurred with the active substance clopyralid, where the toxicity increased over time. For clopyralid the IC_50_ value observed after 15 min of exposure was 10,417 μg·L^−1^. After 30 min, the value was significantly lower 5071 μg·L^−1^. After 15 min of exposure, a higher toxicity of Lontrel 300 was observed compared to the active substance, which may be due to the presence of other compounds in the product. One of these substances is alkylphenol alkoxylate, which is classified according to the CLP regulation as toxic to aquatic organisms with long-term effects [96]. The results are approximately correlated with previously published study values. When the bacteria were exposed to a clear 40,000 μg·L^−1^ solution of clopyralid, a lower bioluminescence inhibition of 83% was measured after 15 min of exposure compared with the control [101]. This can be explained by subsequent degradation via ultraviolet radiation or photocatalytic oxidation, where the degradation products are more toxic [101].

After 15 min of exposure, the IC_50_ value of Finalsan^®^ was set at 64 μg·L^−1^ and after 30 min of exposure, the IC_50_ did not differ very much and remained very low at 66 μg·L^−1^. These results show that of all the evaluated herbicides ever, Finalsan^®^ is the most toxic to *A. fischeri* and Finalsan^®^ can be considered highly toxic to *A. fischeri* bacteria. For use in herbicidal compositions, nonanoic acid is chemically synthesized and mixed with other excipients to improve the properties of the herbicide. One of these substances contained in Finalsan^®^ is propan-2-ol. It is not classified as toxic to aquatic organisms according to CLP, but causes, for example, eye irritation [96]. Compared to pure nonanoic acid, the toxicity of Finalsan^®^ is significantly higher. Half maximal inhibitory concentration for nonanoic acid was 16,040 μg·L^−1^ after 15 min and 14,039 μg·L^−1^ after 30 min from measuring start. This indicates that of all the active substances tested, nonanoic acid causes the lowest inhibition of *A. fischeri* bioluminescence after 30 min. After 15 min, the values approach the dicamba values. At the same time, the results show that there is the biggest difference in toxicity between the herbicide and its active substance. Jones et al. reported nonanoic acid toxicity for *A. fischeri* LC_50_ = 0.360 mM [102], apart from this record, we are not aware of any other results of toxicity testing of the herbicide Finalsan^®^ or nonanoic acid on *A. fischeri*. Significant differences were determined at toxicities of herbicides and also at active substances. The comparison of toxicities of the herbicides are shown in Figure 1 and Figure 2.

### 3.2. Crustacea Bioassay Test

Table 2 shows the LC_50_ and LC_90_ values for all tested herbicides and active substances. The median lethal concentration for Roundup^®^ Classic Pro was determined to be 18 μg·L^−1^; among the tested herbicides and regarding active substances it was ranked the most toxic for *A. salina*. According to de Brito Rodrigues et al., the LC_50_ value for the Roundup^®^ herbicide was determined to be 14 mg·L^−1^ after 48 h of exposure [103]. At a concentration of 28 μg·L^−1^, 90% mortality of the tested organisms was observed. The same concentration of Roundup^®^ converted to glyphosate, i.e., 28 μg·L^−1^ caused 50% mortality of test organisms after 24 h, and after 48 h the LC_50_ value was 19 μg·L^−1^ [104]. Our results show that Kaput^®^ Premium herbicide has very similar LC_50_ value as Roundup^®^ Classic Pro, namely 19 μg·L^−1^. LC_50_ of glyphosate, the active substances of these herbicides, was set to be 811 μg·L^−1^. Of the active substances tested, glyphosate was evaluated as the most toxic to *A. salina*, which corresponds to the conclusions obtained from the toxicity evaluation of the herbicides containing it. Glyphosate and AMPA were found in different concentrations ranging between 0.04 to 700 μg·L^−1^ in a surface water [105,106,107,108,109,110] and between 52 to 3294 μg·kg^−1^ in soil [105,107,108]. Toxicity of glyphosate and its formulations have impact even at environmentally relevant concentrations. Developmental toxicity of 1 μg·L^−1^ of the active substance was observed in juvenile rainbow trout [111], decreased motility and survival of sperm cells of *Astyanax lacustris* was evaluated after exposure to glyphosate-based herbicides at concentrations above 50 ug.l^−1^ [112]. Genotoxicity of Roundup and glyphosate was observed in blood cells of fish at concentrations of 116 and 35.7 ug·L^−1^, respectively [113]. Roundup and glyphosate may have an effect on antioxidant disruption and induce oxidative stress in rats [114,115]. Release of glyphosate into natural habitat may have long-term consequences because of its genotoxic, mutagenic, and hepatotoxic potential in amphibian population associated with agricultural areas [116]. Despite the fact that expected environmental concentration may be lower, environmental factors such as pH or suspended sediment may increase the toxicity of glyphosate-based formulations [117]. In addition, glyphosate can negatively affect the gut of honey bees due to a reduced amount of beneficial bacteria [118]. In the case of young honey bees, a lower sensitivity to a nectar and another changes in their appetitive behavior even in recommended doses of glyphosate were observed. Measured environmental concentration of glyphosate range between 1.4–7.6 mg·L^−1^ [119]. Changes in behavior were observed also at recommended concentrations of commercially formulated glyphosate [120].

Median lethal concentration for Banvel^®^ 480 S was determined to be 2519 μg·L^−1^; compared to *A. fischeri* bacteria, the toxic effect was observed at very similar concentrations. At the same time, Banvel^®^ 480 S herbicide was the least toxic for *A. salina* of all the herbicides tested. The acute toxicity of dicamba in the pure state in tests on *A. salina* was observed at concentrations from 2000 μg·L^−1^, where mortality ranged between 10% and 20%. The concentration that caused the death of 50% of the tested organisms was found to be 4503 μg·L^−1^. Dicamba as emulsifiable herbicide is considered more toxic than its soluble form in *Scinax nasicus* and *Elachistocleis bicolor* [121]. In the case of *Coleomegilla maculata* as an important beneficial insect, a significant decrease in sex ratio after exposing the larvae to the formulation caused by the inert ingredients was observed [30]. Dicamba was found in environment in various concentrations ranging between 0.016 to 0.17 μg·L^−1^ [109]. Long-term genotoxic effects such as primary DNA lesions were observed in *Cnesterodon decemmaculatus* after the exposure to dicamba-based Banvel at 7 and 14 days at a concentration of 41 mg·L^−1^ [122]. Other genotoxic effects such as cellular damage were observed in Chinese hamster ovary cells in a concentration of 500 μg·mL^−1^ of dicamba and Banvel [123]. Hepatic changes with upregulation of the peroxisome proliferator-activated receptor were observed in concentrations of 1% dietary dicamba in both female and male rats, and with lower concentrations up to 0.1% no significant effect was observed [124]. A potential of endocrine disruption such as spermatogenesis inhibition or increased and decreased sex hormone levels was observed on *Gobiocypris rarus* in environmentally relevant concentrations up to 50 μg·L^−1^ [57].

The LC_50_ for Lontrel 300 was determined to be 1796 μg·L^−1^. Median lethal concentration for clopyralid was set to 2800 μg·L^−1^. Chronic effects of clopyralid were evaluated in several organisms. No mortality was observed even at the concentration of 273 mg·L^−1^ in *Oncorhynchus mykiss* [125]. No teratogenic effects in rats or rabbits were observed even at oral doses of 250 mg.kg^−1^.d^−1^ during the major organogenesis [126]. The toxicity of clopyralid, compared to estimated concentrations, poses little or no risk to salmonids fish [125,127], even at the early stages [128] and poses low risk to fish and invertebrates if used as recommended (on grass surfaces) [9]. With a direct overspray to a 2-m-deep pond, the estimated exposure is 55 μg·L^−1^ [125] in storm water runoff from a field was a measured concentration of clopyralid 30 μg·L^−1^ [106].

The concentration causing the death of 50% of the tested organisms was determined to be 100 μg·L^−1^ for herbicide Finalsan^®^ and 7493 μg·L^−1^ for nonanoic acid. A significant increase in toxicity was observed between 70 and 100 μg·L^−1^. This result can be caused, for example, by increasing the amount of other substances that have a negative effect on crustaceans. Among the tested active substances, nonanoic acid was evaluated as the least toxic for *A. salina*. At the same time, when it comes to comparing the toxicity of a herbicide and its active substance, Finlasan^®^ and nonanoic acid have the largest difference in toxicity, where Finlasan^®^ is more than 74 times more toxic than nonanoic acid. Significant differences were determinate at toxicities of herbicides and also at active substances. The comparison of toxicities of the herbicides is shown in Figure 3.

Plant protection products may be found in the environment in greater concentrations than expected or allowed, which may have various effects on microbial communities [129]. Mixtures of plant protection products can be found in various aquatic ecosystems, in which they may have different joint effects compared to effects of individual substances [130]. A stimulatory effect on primary production in the nutrient-deficient freshwater wetland was observed after treatment with a mixture of herbicides at a recommended field application rate (clopyralid as Lontrel, dicamba as Oracle, glyphosate as Glyphos, etc.); similar effects were not observed in the nutrient-deficient saline wetland and changes in the composition of bacterial biofilm were indicated [129].

Regarding the hazard assessment, pesticides are well monitored by local regulation agencies such European Chemicals Agency (ECHA) in European Union or United States Environmental Protection Agency. In the EU, the candidate list was made where the substances of very high concern (SVHC) are listed. None of our active substances has been mentioned in candidate list, so there should be no risk of carcinogenicity, mutagenicity or toxicity for reproduction [131]. However, the EU authorities keep updating data about toxicity and, for example, glyphosate is one of the most observed active substances on the SVHC list. In October 2016, European Food Safety Authority (EFSA) started working on a new risk assessment for glyphosate, where its endocrine activity potential should be evaluated. The conclusion that glyphosate does not have any significant endocrine disruption potential was given by the EFSA review from August 2017 [132]. In every review of the pesticide risk assessment, the conclusion (RPAC) has a second part that reflects the ecotoxicity effect. Based on all risks, the active compound is approved or not approved or approved with some notice. All of them are approved for use in agriculture and horticulture by the European commission, the five year approval is issued until December 2022 for glyphosate and dicamba, April 2022 for clopyralid, and August 2022 for nonanoic acid [133]. Except for nonanoic acid, these active substances have some side effects, which are known and should be mentioned, for example acute toxicity and aquatic chronic toxicity caused by dicamba and aquatic chronic toxicity caused by glyphosate [96]. Unfortunately, data of experiments with Aliivibrio fischeri or Artemia salina are not mentioned in RPAC. However, there are tests on *Daphnia magna* and larvae of *Neohelice granulata* for glyphosate and *Chironomus riparius* for glyphosate acid [134].

## 4. Conclusions

The measured results show that for the marine bioluminescent bacteria *A. fischeri*, the tested herbicides and their active substances have the highest inhibitory abilities in the following order: Finalsan^®^ > Roundup^®^ Classic Pro > Banvel^®^ 480 S > Kaput^®^ Premium > glyphosate > Lontrel 300 > clopyralid > nonanoic acid > dicamba. The order of toxicity effect for *A. salina* crustaceans is as follows: Roundup^®^ Classic Pro > Kaput^®^ Premium > Finalsan^®^ > glyphosate > Lontrel 300 > Banvel^®^ 480 S > clopyralid > dicamba > nonanoic acid. The results indicate acute toxicity for *A. fischeri* and *A. salina*. The authors suggest that more attention should be paid in the future to the chronic toxicity and combined effects of active substances and pesticides.

## Figures and Tables

**Figure 1 toxics-09-00275-f001:**
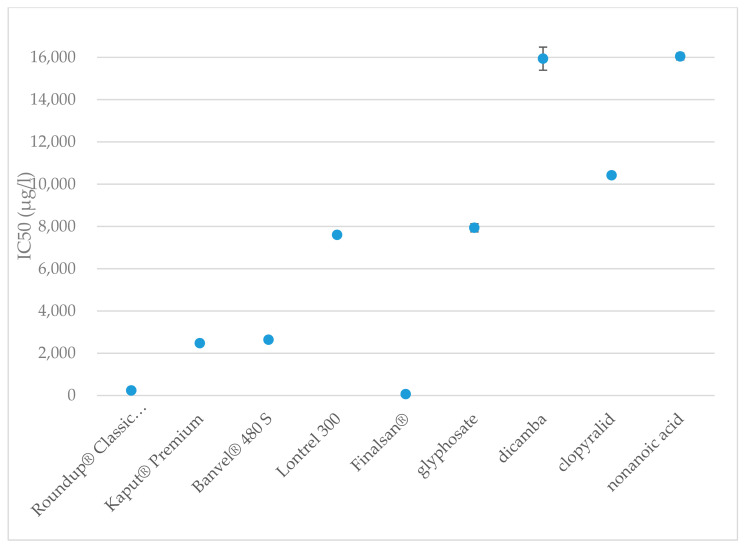
Toxicities of the herbicides and active substances after 15 min of exposure to *A. fischeri*. The data are expressed as the mean IC_50_ value ± their respective standard deviations.

**Figure 2 toxics-09-00275-f002:**
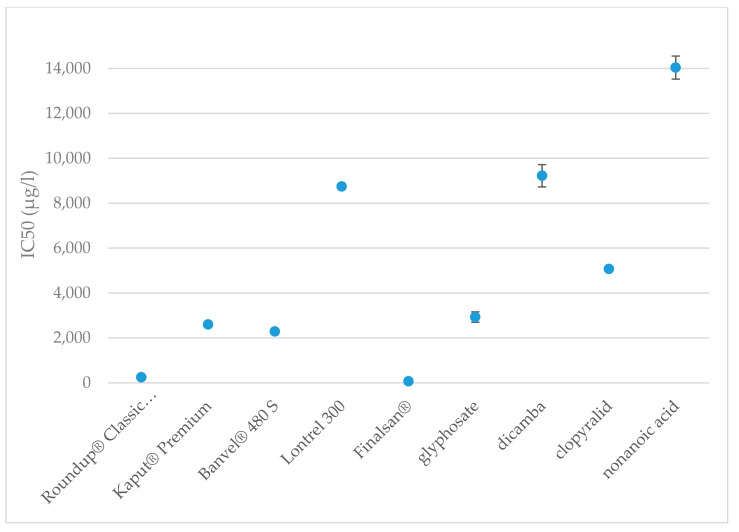
Toxicities of the herbicides and active substances after 30 min of exposure to *A. fischeri*. The data are expressed as the mean IC_50_ value ± their respective standard deviations.

**Figure 3 toxics-09-00275-f003:**
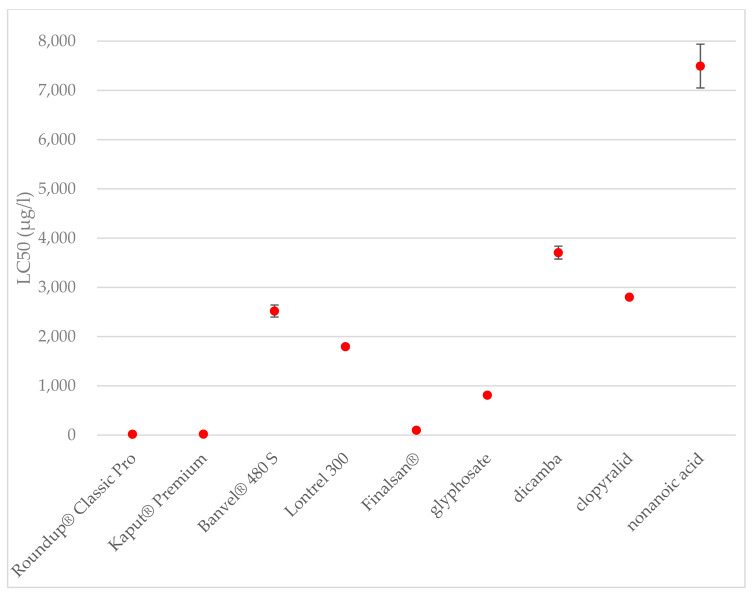
Toxicities of the herbicides and active substances after 24 h of exposure to *A. salina*. The data are expressed as the mean LC_50_ value ± their respective standard deviations.

**Table 1 toxics-09-00275-t001:** IC_50_ and IC_90_ values of all tested herbicides and active substances, and 95% fiducial confidence intervals at 0.05 level of significance, tested organism *A. fischeri*.

Tested Substances	Time (min)	IC Values					
		IC_50_ ( μg·L^−1^)	95% Fiducial CI		IC_90_ ( μg·L^−1^)	95% Fiducial CI	
			Lower	Upper		Lower	Upper
Roundup^®^ Classic Pro	15	236	194	288	494	405	602
	30	243	203	290	467	391	557
Kaput^®^ Premium	15	2475	1887	3247	7438	5670	9757
	30	2598	2005	3368	7400	5709	9591
Banvel^®^ 480 S	15	2637	1859	3740	14,469	10,201	20,521
	30	2286	1624	3217	11,748	8348	16,532
Lontrel 300	15	7596	5023	11,488	38,749	25,623	58,600
	30	8740	6017	12,696	37,188	25,601	54,019
Finalsan^®^	15	64	31	130	1096	542	2216
	30	66	30	144	1567	723	3395
Glyphosate	15	7934	3836	16,410	145,009	70,116	299,896
	30	2928	1421	6033	42,507	20,633	87,570
Dicamba	15	15,937	10,267	24,738	87,040	56,073	135,109
	30	9220	5558	15,296	63,817	38,468	105,868
Clopyralid	15	10,417	6685	16,231	58,985	37,855	91,910
	30	5071	3107	8276	31,126	19,072	50,800
Nonanoic acid	15	16,040	8105	31,745	228,878	115,649	452,965
	30	14,039	7,444	26,476	163,374	86,631	308,100

IC_50(90)_: inhibition concentration of tested substances that caused the inhibition of 50% (90%) of exposed bacteria (probit analysis), 95% fiducial CI: confidence interval.

**Table 2 toxics-09-00275-t002:** LC_50_ and LC_90_ values of all tested herbicides and active substances and confidence intervals at 0.05 level of significance, tested organism *A. salina*.

Tested Substances	LC Values					
	LC_50_ ( μg·L^−1^)	95% Fiducial CI		LC_90_ ( μg·L^−1^)	95% Fiducial CI	
		Lower	Upper		Lower	Upper
Roundup^®^ Classic Pro	18	16	21	28	24	32
Kaput^®^ Premium	19	16	23	31	26	36
Banvel^®^ 480 S	2519	2083	3046	4669	3861	5647
Lontrel 300	1796	1304	2473	5776	4194	7955
Finalsan^®^	100	59	169	649	383	1100
Glyphosate	811	729	902	1060	952	1179
Dicamba	3705	2793	4914	8628	6505	11,444
Clopyralid	2800	2565	3056	3797	3478	4144
Nonanoic acid	7493	6568	8549	12,516	10,971	14,278

LC_50(90)_: lethal concentration of tested substances that killed 50% (90%) of exposed organisms (probit analysis), 95% fiducial CI: confidence interval.

## Data Availability

Not applicable.

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
