# Peer review of "The Influence of Herbicides to Marine Organisms Aliivibrio fischeri and Artemia salina"

_toxics, 2021, doi:10.3390/toxics9110275_

Round 1

Reviewer 1 Report

Line 31: the authors should talk about herbicide run-off, e.g. drift – how it happens, and what are the consequences

Line 33: not only solubility. List all factors influence herbicide drift

Line 101: explanations needed “IC50 and IC90 values”

Line 102: not clear “herbicides and their active substances”: roundup is glyphosate; banvel is dicamba. This is a way strange. Kaput premium is also glyphosate. You must mentioned that you have used the same active ingredient, while different products

Line 111- 120: I would suggest putting this information in a table

It must be a reason why those herbicides were chosen? Provide more literature regarding this issue.

Glyphosate is the most used herbicide worldwide (ref…); Adoption of glyphosate tolerant crops, dicamba tolerant crops…

Line 198: here you have pure glyphosate; I have not seen this information in M&M

IC does not have the explanation in the text. Oh here it is in the line of the 214

Lines 208-211: this part of discussion is to short. Put more effort to compare your findings with other published literature

Line 222: remove “on the other hand”

The discussion part must be better connected with published literature

Line 314: and what should we do regarding this issue?

Author Response

Thanks for your feedback, we appreciate it.

Line 31: We have added information on herbicide run-off.

Line 33: We have listed other factors.

Line101: We explained the values.

Line 102: We mentioned that we have two herbicides with the same active ingredient.

Line 111 – 120: If you don't mind, we would like to leave it in the text.

Why choosen herbicide - We added information that these are the best-selling (most used) herbicides in the Czech Republic and added references.

Line 198: We removed the word "pure".

Lines 208-211: Yes, the discussion was short. We have expanded it with additional information, the effect of herbicides on other organisms, chronic toxicity and REACH.

Line 222: Deleted.

Reviewer 2 Report

The manuscript contain new important data, well in line with “Toxics” aims, however the weak  point is the discussion. The discussion in this manuscript is rather modest so the authors are asked to improve this part (different aspects can be taken into account: environmental consequences (taken into account environmental concentrations), with special focus to marine/coastal ecosystems (taken into account the measured concentrations, discussion with other papers see e.g. The Effects of Glyphosate and Its Commercial Formulations to Marine Invertebrates: A Review” by V.Matozzo, J.Fabrello and M. Marin in J. Mar. Sci. Eng. 2020 and papers cited there), comparison with other ecotoxicity papers published in this field, suggestions for future studies (concentrations, chronic tests?, test organisms etc....... At the present stage it is rather a report (however it should be underlined that the presented data are of high value) not a manuscript which can be accepted without improvements in journal with important impact factor

More detailed suggestions for some minor improvements are presented below

Line 26/27 „How herbicides affect the living also on their ability to move within it” something is missing in this sentence

Line 27 “The most important factors influencing their movement include..” I suggest to replace movement by distribution, it seems to be more relevant

Line 39, reference 11 – there is more recent reference concerning the Baltic Sea: Chemosphere 2020, “The challenge of detecting the herbicide glyphosate and its metabolite AMPA in seawater e Method development and application in the Baltic Sea” by Marisa A. Wirth, Detlef E. Schulz-Bull, Marion Kanwischer

Line 63 “M. Aeruginisa” full name should be used when appearing for the first time in the text, also correct aeruginisa to aeruginosa

Line 65 change “Allivibrio” to “Aliivibrio”

Line 73-80 “According to Tsiridis et al., the toxicity of copper to A. fischeri decreases with

the addition of humic acids (HA), ........ rate of cell death is similar to bioluminescence inhibition [37].

This part is not necessary I mean it is not relevant to the aim of this study so should be removed

The same remark is valuable for part from line 88 to 96 “Significant changes in the hatchability, ....... the frequency of apoptotic cells, and DNA damage in nauplii in direct correlation with increasing concentrations of the toxicant [50].”

Instead it would be better to show the aspect of novelty of the presented manuscript and its importance (even if “only” acute tests were applied

Line 140-141 – “The test substances were diluted in sodium chloride solution and measured .... I guess the concentrations were measured, please explain which method was used to measure the analyte concentrations, please explain also what was the criterion for choosing the applied concentration levels?

materials and methods: please explain briefly in the text what was used as control in both tests, did the control “take into account” DMSO? Of course, I have noticed that the authors tested DMSO influence on used organisms prior to its use with the test substances but the most reliable way would be the use of the same amount of DMSO in control samples as in the samples with analytes, quite possible that the experiment was done in that way but

toxicity

line 201 “Some excipients may increase the effects of pesticides, so it is appropriate to perform toxicity tests on a formulation, such as Roundup, rather than on the pure substance itself” This statement is absolutely true, but of course the things are even more complicated e.g. both active substances and excipients may degrade before they reach marine ecosystem, may I ask the authors to comment on this issue in the text (e.g. in the discussion)

line 262 “Significant differences were determinate at toxicities of herbicides and also at active substances “determinate” – rather determined? Please check this sentence for language correctness

 control level- how it was taken into account

line 266” Figure 1. Significant differences in toxicities of the herbicides and active substances after 15 min of exposure to A. fischeri. .....” It would be more appropriate to change the caption to “Differences....” in the present form it is hard to guess which values differ significantly – see e.g Kaputt Premium and glyphosate. The same remark is valuable for figure 2 and 3 captions.

Line 274-284 – something went wrong with italics, the same for lines 290-317.

Line 304 “comparasion”

Author Response

Thank you for your time in evaluating our article.

We have expanded the introduction and discussion.

In the article we have cited both articles, which mention review and are related to the organisms we tested.

Line 26/27: We rewrote the sentence, thank you for noticing the mistake.

Line 27: Changed, thank you for the improvement.

Line 39: Thanks for pointing out an interesting article. We added information from it to our article

Line 63: Of course, fixed.

Line 65: Changed, thanks.

Line 73-80: We removed the sentences. We added information to the discussion about the importance of monitoring toxic substances.

Line 140-141: Information added.

Materials and Methods: We have added a section on DMSO to the materials and discussion.

Line 201: We have added information about AMPA to introduction.

Line 262: Checked.

Line 266: We've changed the label and graphs.

Line 274-284, 290-317: Yes, we apologize. We have fixed this error.

Line 304: Corrected.

Reviewer 3 Report

The authors studied the acute toxicity of herbicide formulations and of the corresponding active substances on the luminescent bacteria Aliivibrio fischeri and nauplii of the crustacea Artemia salina. These models are marine organisms, traditionally used in biomonitoring for aquatic toxicity screening.

The authors showed that the acute toxicity of the active substances on the two marine models was lower than the toxicity of the corresponding formulations by one or two orders of magnitude. This indicates that the components of the formulations and/or the chemicals combined to the active substances or the additive surfactants added are not neutral and increase the toxicity of the active substance. This is not surprising and not new. However the demonstration is clear here.

The introduction of the paper is well documented, but some details are not useful in the work : for instance, AMPA is reported as the main metabolite of glyphosate and is said to impact non-target organisms in the environment, but AMPA is not tested in this work. Also, the information given is not discussed further : as an example, the mode of action of the synthetic auxin type dicamba may explain why dicamba is poorly toxic to the bacteria and the crustacean, which are not biological model relevant for this mechanism of toxicity.

In the materials and methods, the authors should mention that the luminescent bacteria are tested in a saline solution (2%) less salted than the 3% salted solution used for Artemia. This difference in salt concentrations may influence the intensity of the bacterial response.

The comments of the results are clear, but there are problems in the figures that do not reflect the data presented in the tables. There are mistakes either in the tables or in the corresponding figures : the IC50s values of glyphosate, Banvel® 480S, dicamba, clopyralid, Finalsan® on A. fischeri, which are reported in figures 1  and 2, do not correspond to the values presented in table 1. The same problem is found with the LC50s of these chemicals on A. salina in figure 3 and table 3.

In tables 1 and 2, the title of the tables should indicate that the reported results concern the bacteria A. fischeri. Also the unit (µg/L) of the concentrations should be indicated besides “IC50 values” in the table and not in a footnote. Same remark for table 3 regarding LC50 values for nauplii of Artemia salina.

To our opinion, tables 2 and 4 are not useful and should be deleted from the main text (at least shifted in supplementary materials).

The discussion is relatively short.  The authors compare the rank of toxicity of the tested chemicals on each model. It could have been of interest to discuss the significance of the outcomes (i.e. the higher acute toxicity of formulations compared to the toxicity of pure active substances : what about chronic toxicity ?) in an environmental point of view and in the long term.

 Also, a discussion on hazard assessment of active substances and formulations in a regulatory context might have been of interest ; also about chronic toxicity of phytosanitary formulations.

Minor remarks : 1) line 300  "substances" has to be corrected. 2) a part of section 3 (Results and discussion) and 4 (conclusion) is in italic : why ?

Author Response

Thanks for your feedback, we appreciate it.

In the introduction, we provided information about the synthetic auxin.

We have added information about the salinity of the solutions to the Materials and Methods section. The salinity of the solutions for luminescent bacteria was selected according to the standard.

As for the figures in the results section, thank you for noticing the mistakes. It was fixed.

We added the names of the tested organisms to the names of the tables. Units were added directly to the table.

Regarding Tables 2 and 4, we provided this information (statistics) specifically at the request of the academic editor. If it suits the editors, we do not oppose moving them to the supplementary materials section.

The discussion has been reworked. We've added a discussion about chronic toxicity.

Minor remarks:

  • the word has been corrected to "substances", thanks for noticing
  • During the final recording, the font was inadvertently changed to italic.We apologize and, of course, there has been a correction.

Round 2

Reviewer 1 Report

N/A

Author Response

The spelling errors was fixed and the tables 2 and 4 have been moved to the Supplementary Materials.

Reviewer 2 Report

-

Author Response

(The authors gave the same response as above.)

Reviewer 3 Report

The manuscript can still be improved : a few spelling errors have to be corrected (see below). Also the expression of IC50 and IC90 values (in µg/L) should be given without decimals or centesimals because the confidence interval is of the order of several or hundreds of µg/L. Then, the decimals and centesimals should be deleted in the values of the tables 1 and 3 and in the text : such a precision has no sense on a statistic point of view. For instance in table1, the gives for "Round up Classic Pro IC50 (95% CI): 15 minIC50 = 236 (194-288) µg/L ; 30minIC50 = 243 (204-290) µg/L"

The same cleaning has to be done for Artemia salina LC50 values in table 3.

The figures 1-4 are presenting the IC50 or LC50 values expressing the toxicity of the herbicides and not the differences in toxicities. Therefore "Differences in" has to be deleted : the legend can be : Toxicities of the herbicides... . The data is expressed (not expresed)...

The tables 2 and 4 are intermediary results leading to the determination of the toxicity values. They are useless in the main manuscript and can be either omitted (or shifted in supplementary data).

line 63 "than" instead of "then"; line 246 : "luminescent"

Author Response

Thanks for the further recommendations. We have modified our manuscript. We fixed spelling errors. As for the decimals in the resulting IC and LC values, these were adjusted both in the text and in the tables. Tables 2 and 4 have been moved to supplementary materials.